# GAGE is a method for identification of plant species based on whole genome analysis and genome editing

Lijun Hao[1,5], Wenjie Xu[1,5], Guihong Qi[1], Tianyi Xin[1], Zhichao Xu[1], Hetian Lei[2] & Jingyuan Song [1,3,4 ✉]

Whole genomes of plants should be ideal databases for their species identification, but unfortunately there was no such method before this exploration. Here we report a plant species identification method based on the whole Genome Analysis and Genome Editing (GAGE). GAGE searches for target sequences from the whole genome of the subject plant and specifically detects them by employing a CRISPR/Cas12a system. Similar to how Mendel chose *Pisum sativum* (pea), we selected *Crocus sativus* (saffron) to establish GAGE, in which we constructed a library containing all candidate target sequences. Taking a target sequence in the ITS2 region as an example, we confirmed the feasibility, specificity, and sensitivity of GAGE. Consequently, we succeeded in not only using GAGE to identify *Cr. sativus* and its adulterants, but also executing GAGE in the plants from different classes including angiosperms, gymnosperms, ferns, and lycophytes. This sensitive and rapid method is the first plant species identification method based on the whole genome and provides new insights into the application of the whole genome in species identification.

[1] Key Lab of Chinese Medicine Resources Conservation, State Administration of Traditional Chinese Medicine of the People's Republic of China, Institute of Medicinal Plant Development, Chinese Academy of Medical Sciences & Peking Union Medical College, Beijing, China. [2] Shenzhen Eye Hospital, Shenzhen Eye Institute, Jinan University, Shenzhen, China. [3] Engineering Research Center of Chinese Medicine Resource, Ministry of Education, Beijing, China. [4] Yunnan Key Laboratory of Southern Medicine Utilization, Yunnan Branch Institute of Medicinal Plant Development, Chinese Academy of Medical Sciences, Jinghong, China. [5] These authors contributed equally: Lijun Hao, Wenjie Xu. ✉email: jysong@implad.ac.cn

Species identification arose with man's first exploration of nature and laid an important foundation for life sciences. In plant research, plant species identification provides basic data for ecology, botany, evolutionary biology, and other flora-based disciplines. Despite its emerging thousands of years ago, it is still one of the most vibrant research fields, introducing and incorporating many frontier strategies and technologies[1]. In early research, plants were described and classified into different groups mainly based on their morphological features[2]. Subsequently, introduction of novel technologies and strategies such as microscopy and spectroscopy advanced the development of anatomic taxonomy and chemical taxonomy[3,4].

In the 20th century, as DNA sequencing data was collected and molecular identification methods such as DNA barcoding arose[5–7], plant species identification was fully revolutionized. As the carrier of genetic information, the whole genome has been recognized as the ideal database for plant species identification, considering its abundant information. Thus, plant species identification based on whole genome analysis is very promising. Previous utilization of whole genome in identification of plant species was restricted due to a shortage of published genomes and a huge demand for computer resources, but now DNA sequencing is convenient and affordable, and techniques for genome analysis have been enormously improved[8,9]. According to statistics, more than 700 plant genome sequencing data published up to September 23, 2021[10], providing a firm database for genome analysis. Moreover, the advancement of computer hardware and the development of novel programs such as Bowtie has made the analysis of genomes both faster and more efficient[11,12], indicating that the genome-wide identification age is coming.

Notably, the clustered regularly interspaced short palindromic repeats (CRISPR)/CRISPR-associated proteins (Cas) era has arrived[13] and the CRISPR/Cas12a system has been introduced into the identification of virus and bacteria[14–16]. The combination of this frontier strategy of genome editing and the whole genome analysis is opening a new dimension to plant species identification. Herein we describe a method for plant species identification based on Genome Analysis and Genome Editing (GAGE), which has capacity in accurately identifying the subject plant species from the adulterants. GAGE searches for the target sequence from the whole genome and specifically detects it by introducing CRISPR/Cas12a system.

GAGE includes two key steps: bioinformatic analysis and experimental evidence, i.e., genome analysis (GA) and genome editing (GE). The first step in the bioinformatic analysis is to screen candidate target sequences with a nearby protospacer adjacent motif (PAM) by analyzing the genome of the plant species. For plant species identification, only target sequences present in the genome of plant species and not in the genomes of its related species are considered for use, which is accomplished by alignment to the genomes of the related species. This process focuses on the target sequences in the intraspecifically conserved, species-specific regions. In the experimental evidence, the CRISPR RNAs (crRNAs) used to guide Cas12a are designed according to the target sequences. As the synthesized crRNAs hybridize to the protospacer of target sequences, the collateral cleavage activity of Cas12a is activated, resulting in the nicking of single-stranded DNA reporter (Poly_A_FQ: 5'-FAM-AAAAAAAAAA-BHQ-3') by the RuvC domain of Cas12a and the subsequent generation of fluorescence (Fig. 1).

## Results

**Construction of target sequences library of *Crocus sativus*.** To establish GAGE, whole genome sequences are necessary. The whole genome of *Crocus sativus* (saffron) was determined in an early study, and considering its high economic and medicinal value, we selected *Cr. sativus* as a demonstration of GAGE. *Cr. sativus*, one of the most expensive herbs in the world, is wildly cultivated in the Mediterranean, east Asia, and Irano-Turanian region for its highly valuable application in medicine, tea, and

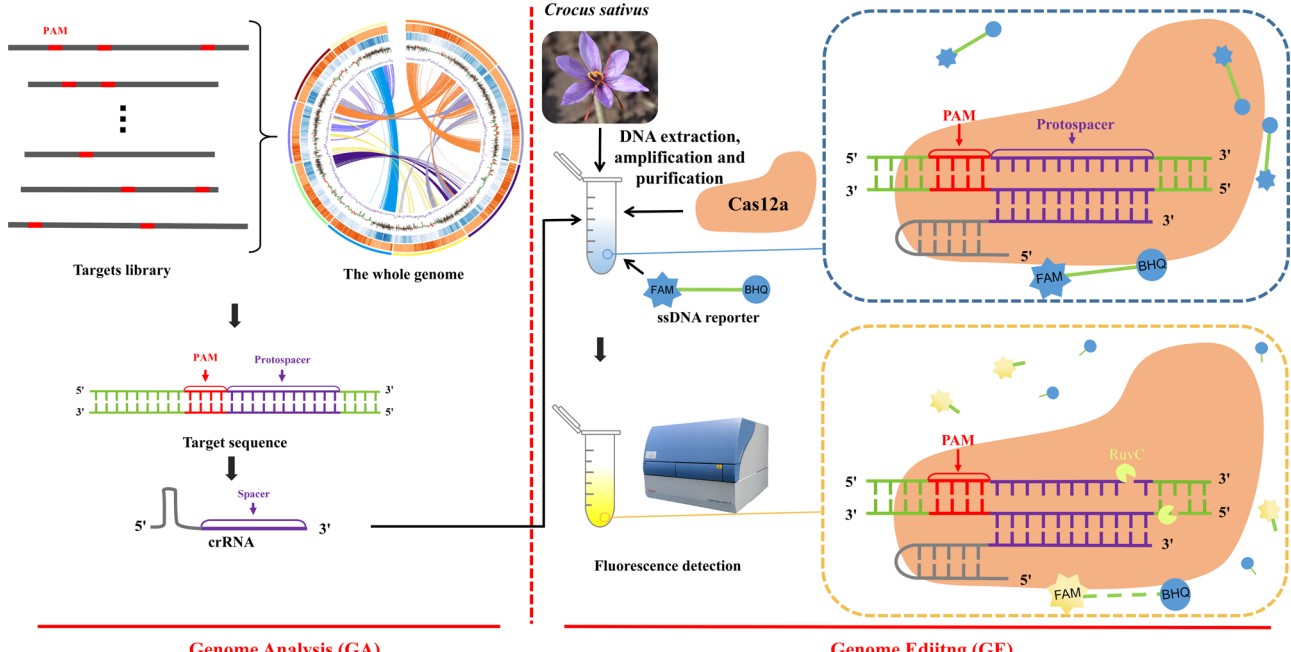

**Fig. 1 The strategy of the whole Genome Analysis and Genome Editing (GAGE).** The fluorophore (6-Carboxyfluorescein, FAM) is linked to the quencher (Black Hole Quencher-1, BHQ) through ten dAs (ssDNA) and is initially quenched. After breaking away from BHQ, FAM absorbs light at its excitation wavelength and emits fluorescence. PAM: protospacer adjacent motif; The whole genome: The schematic genome draft was cited from the reference[32]; Targets library: The gather of target sequences; Target sequence: The sequence with a nearby PAM; crRNA: CRISPR RNA; ssDNA reporter: single-stranded DNA reporter (5'-FAM-AAAAAAAAAA-BHQ-3', Poly_A_FQ).

cooking seasonings for ages[17,18]. We first identified the candidate target sequences (hereinafter referred to as Targets) with a nearby PAM in the genome of *Cr. sativus*. The genome sequence analysis revealed that there were more than 178 million Targets in the genome of *Cr. sativus*, and that nearly one-third of these Targets remained after deduplication (Table 1). On average, there was one potential target sequence per 26.8 bp in the genome of *Cr. sativus*. For all Targets in the annotated regions of the genome, most of them were located in the coding genes; only 21,275 of them were located in non-coding RNAs. Among Targets located in coding genes, 1,997,115 of them were located in DNA coding sequences (CDSs), which could be used for screening Targets in functional genes. The Targets in the intraspecifically conserved, species-specific regions are more likely to be the final target sequence used for identification and the universal DNA barcoding regions had been demonstrated to have these characteristics[5]. In order to rapidly evaluate GAGE, we focused on the Targets in ITS2 region of *Cr. sativus*, although Targets in other regions with the above characteristics also would be suitable, such as the *ycf1* and *leafy* genes[19,20]. There was only one target sequence which we named Cs_target1 (Fig. 2a), located in ITS2 region, that had 201 copies in the genome (Fig. 2b). The matched crRNA of Cs_target1 (Cs_crRNA) was designed by adding crRNA repeat to the upstream of spacer (Fig. 2a).

**Feasibility, specificity, and sensitivity of GAGE.** Taking Cs_target1 as crRNA, ITS2 fragments of *Cr. sativus* as DNA substrate, we confirmed the feasibility, specificity, and sensitivity of GAGE. We first investigated the endonuclease activity and collateral cleavage activity of LbaCas12a to test the feasibility of GAGE. As shown in Fig. 3a, the complex of Cas12a and Cs_crRNA significantly cleaved the DNA substrate and generated short DNA fragments. In Fig. 3b, b1, b2, b3 had the same 50 bp ssDNA (Fig. S1), but only the ssDNA in b1 was digested by the powerful collateral cleavage activity of Cas12 coupled with DNA substrate and crRNA. Encouraged by this, we next assembled the reaction with a ssDNA reporter (Poly_A_FQ: 5'-FAM-AAAAAAAAAA-BHQ-3') to finally evaluate the feasibility of GAGE and successfully detected fluorescence signal (Fig. 3c). So we believed that GAGE is a feasible approach.

The specificity of GAGE was shown in Fig. 3d, the two groups had the same complex expected for DNA substrate. Only when the ITS2 fragments of *Cr. sativus* were present, Cas12a digested the Poly_A_FQ and generated fluorescence signal, which rose rapidly in a short time and was significantly higher than that of the negative control (CK). The result provided a firm support for the specificity of GAGE. In order to determine the sensitivity of GAGE, the ITS2 fragments of *Cr. sativus* were diluted ten-fold to obtain a series of DNA substrates with six final concentrations, ranging from 0 to 10 ng/μL. As shown in Figs. 3e, e2 (1 ng/μL) had the highest fluorescence signal, which reached its maximum in the shortest time. The *t*-test showed that there was no significant difference ($P < 0.01$) between e5 (0.001 ng/μL) and CK during the 35 min, but significant difference ($P < 0.01$) was found between e4 (0.01 ng/μL) and CK, so the limit of detection (LOD) of GAGE was considered as 0.01 ng/μL.

**Identification of *Crocus sativus* with GAGE.** *Cr. sativus* and its adulterants were first subjected to GAGE. The stigma of the *Cr. sativus* is used for medicinal purposes and for perfume material. Because of its high value and low yield, many plant materials of other plant species with similar characteristics, including the flower of *Carthamus tinctorius* (safflower), the stamen of *Nelumbo nucifera* (lotus), and the style and stigma of *Zea mays* (corn), are dyed red and used to impersonate the stigma of *Cr. sativus*[21]. For the identification of *Cr. Sativus* (saffron) and its adulterants, only Targets present in the genome of *Cr. Sativus* and not in the genome of *Ca. tinctorius*, *N. nucifera*, and *Z. mays* were considered as a contender for the final target sequence for identification. In addition, previous studies showed that off targets occur with three or less mismatches[22]. Based on the target sequence, the crRNA was designed to recognize and bind to the DNA substrate from *Cr. Sativus*, which further drove the

| Classification | Total target number | Percentage | Deduplicated target number |
|---|---|---|---|
| Genome | 178,043,117 | 100% | 59,282,259 |
| Unannotated regions | 151,251,456 | 85% | 46,700,437 |
| Annotated regions | 26,791,661 | 15% | 12,581,822 |
| Coding genes | 26,771,965 | | 16,626,193 |
| CDS regions | 1,997,115 | | 1,262,246 |
| Non-coding RNAs | 21,275 | | 5111 |

**Table 1 Statistics of Targets in different classification.**

Total target number: all Targets of *Crocus sativus*; Deduplicated target number: Targets after removing duplicates; Genome: Targets located in the whole genome; Unannotated regions: Targets located in the unannotated regions of genome; Annotated regions: Targets located in the annotated regions of genome; Coding genes: Targets located in coding genes of *Cr. sativus*; CDS: Targets located in CDS of *Cr. sativus*; Non-coding RNAs: Targets located in miRNA, rRNA, snRNA or tRNA.

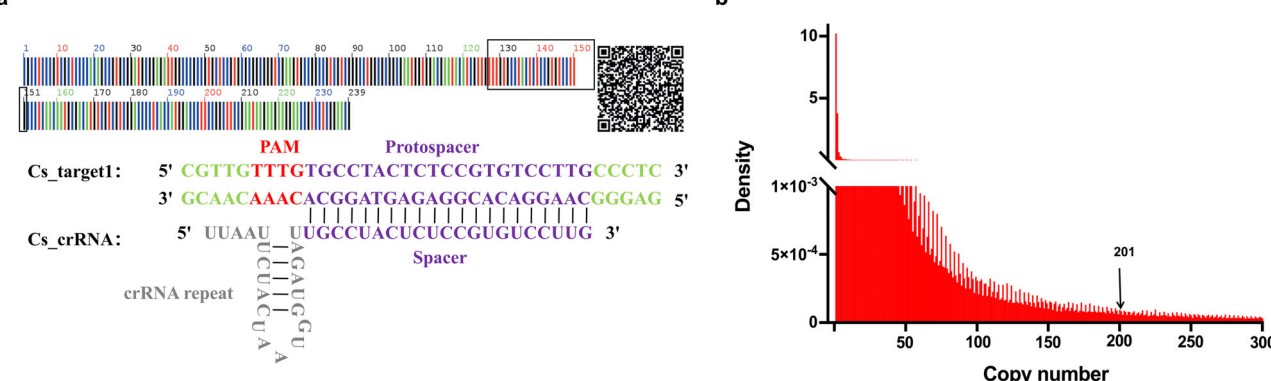

**Fig. 2 Targets in ITS2 region of *Crocus sativus*. a** Target sequence located in ITS2 region of *Crocus sativus*, Cs_target1 and its crRNA, Cs_crRNA. The red sequence is PAM and purple sequence is protospacer in Cs_target1; The gray sequence is the universal sequence of crRNA for LbaCas12a and purple sequence is spacer in Cs_crRNA. **b** Density of Targets' copy numbers ranging from 1 to 300 in *Cr. sativus*. Cs_target1 has 201 copies in the genome (**a**).

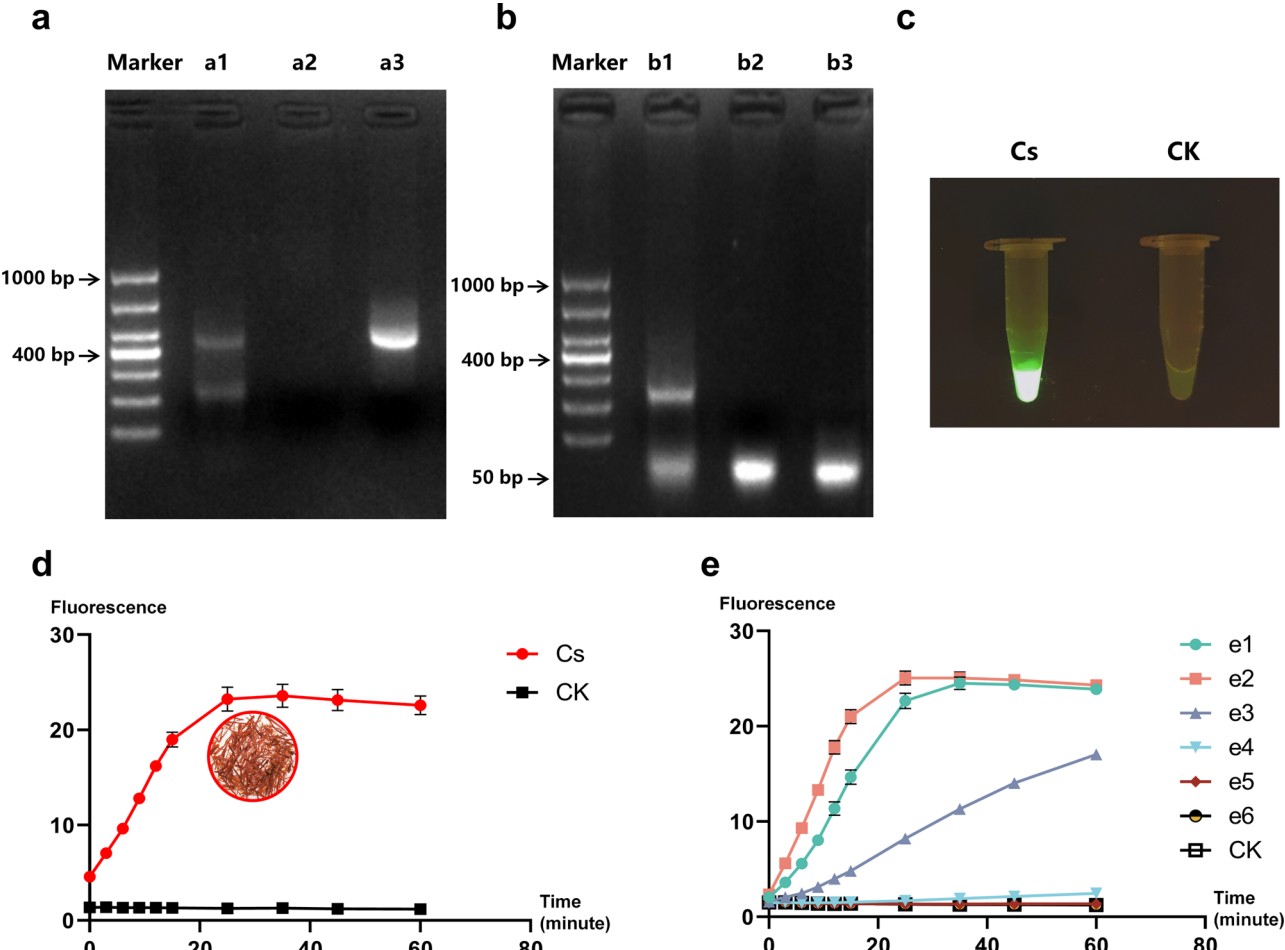

**Fig. 3 Feasibility, specificity and sensitivity of GAGE. a** Endonuclease activity of Cas12a. Lane Marker: DL1000 (Takara Biomedical Technology (Beijing) Co., Ltd., China); Lane a1: Cas12a + Cs_crRNA + ITS2 fragments of *Crocus sativus*; Lane a2: Cas12a + Cs_crRNA: Lane a3: ITS2 fragments of *Cr. Sativus*. **b** Collateral cleavage activity of Cas12a. Lane b1: Cas12a + Cs_crRNA + ITS2 fragments of *Cr. sativus* + 50 bp ssDNA; Lane b2: 50 bp ssDNA; b3: Cas12a + Cs_crRNA + 50 bp ssDNA. **c** Fluorescence signal of GAGE. The two groups were incubated at 37 °C for 25 min and detected by excitation with 470 nm light. Cs: Cas12a + Cs_crRNA + ITS2 fragments of *Cr. sativus* + ssDNA reporter; CK (negative control): Cas12a + Cs_crRNA + H₂O + ssDNA reporter. **d** Specificity of GAGE. The plant materials in the circle are the stigma of saffron. The DNA substrate of two groups were as follows: Cs: ITS2 fragments of *Cr. sativus* and CK (negative control): nuclease-free water. **e** Sensitivity of GAGE. The seven groups had different concentration of DNA substrates (ITS2 fragments of *Cr. sativus*), e1: 10 ng/µL; e2: 1 ng/µL; e3: 0.1 ng/µL; e4: 0.01 ng/µL; e5: 0.001 ng/µL; e6: 0.0001 ng/µL and CK (negative control): 0 ng/µL. All plots data represented means +/− standard deviation (SD) from three independent replicates.

generation of fluorescence. Because there was no sequence in the genomes of adulterants that matched the designed crRNA, no fluorescence was generated when the sequences from adulterants were used as DNA substrate. So we analyzed the specificity and predicted off targets of Cs_target1 by mapping it to the genomes of *Cr. sativus* and its three adulterants. The prediction results showed that there was no sequence within three base mismatches compared to Cs_target1 in the genome of *Ca. tinctorius*, *N. nucifera*, and *Z. mays*, and that there were 13 sequences with one base mismatch and 8 sequences with two base mismatches in the genome of *Cr. sativus* (Fig. 4a and Tables S1, S2), so we chose Cs_crRNA as crRNA to identify *Cr. sativus*. The ITS2 fragments of *Cr. sativus* and its three adulterants were used as DNA substrate, and every group had the same composition except the DNA substrate. As depicted in Fig. 4b, only Cs generated the detectable fluorescence signal, which reached the maximum at 25 min and remained steady until the end of the assay. The groups comprising ITS2 fragments from the adulterants did not generate fluorescence signal. These results indicated that GAGE can specifically identify *Cr. sativus* from its adulterants within a short time.

**Application of GAGE in plants from different classes.** In order to evaluate the applicability of GAGE, it was further subjected to *Ricinus communis*, *Setaria italica*, *Ginkgo biloba*, *Alsophila spinulosa*, and *Selaginella tamariscina*. Among them, *R. communis* and *S.italica* belong to angiosperms; *G. biloba* belongs to gymnosperms; *A. spinulosa* belongs to ferns, and *Sel. tamariscina* belongs to lycophytes. Similar to the procedures in *Cr. sativus*, we first analyzed the genomes of *R. communis*, *Set. italica*, *A. spinulosa* and *Sel. tamariscina*, there was one target sequence per 18.5 bp, 29.9 bp, 26.3 bp, and 24.6 bp, covering 99.97%, 94.14%, 99.94%, and 99.91% of coding genes (CDS regions), respectively (Fig. 5a). The Targets located in ITS2 region of the four species were also extracted: there were three targets in both *R. communis* and *Set. italica*, two targets in *A. spinulosa* and only one in *Sel. tamariscina*. The copy number of all three targets of the *R. communis* genome was more than 200, but there were only 9–23 copies of the targets in *Sel. tamariscina*, *Set. italica*, and *A. spinulosa* (Table S3). Because the full genome size of *G. biloba* is too big to analyze, we constructed a smaller Targets library in its ITS2 region rather than the whole genome and extracted two targets (Gb_target1 and Gb_target2). Finally, GAGE was

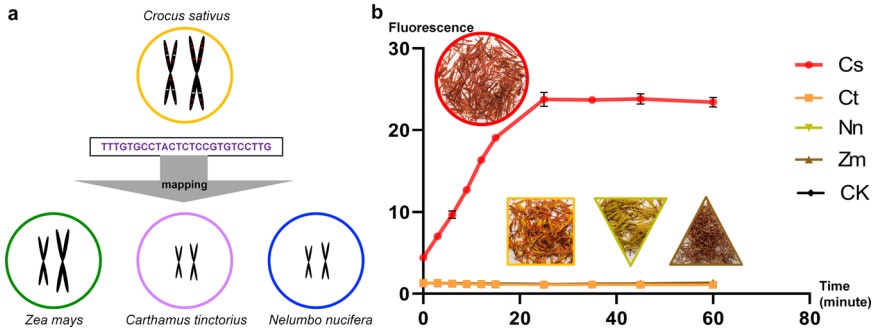

**Fig. 4 Identification of *Crocus sativus* with GAGE. a** Selection of target sequences. The red lines represented Cs_target1 and its copies and white lines represented the sequences within three base mismatches compared to Cs_target1. **b** Results of the identification of *Cr. sativus* with GAGE. The plant materials in the circle, square, triangle, and inverse triangle are the stigma of saffron, the flower of safflower, the stamen of lotus, and the style and stigma of corn, respectively. The DNA substrates of each group were as follows: Cs: ITS2 fragments of *Cr. sativus*; Ct: ITS2 fragments of *Ca. tinctorius*; Nn: ITS2 fragments of *N. nucifera*; Zm: ITS2 fragments of *Z. mays* and CK (negative control): nuclease-free water. All plots data represent means +/− standard deviation (SD) from three independent replicates.

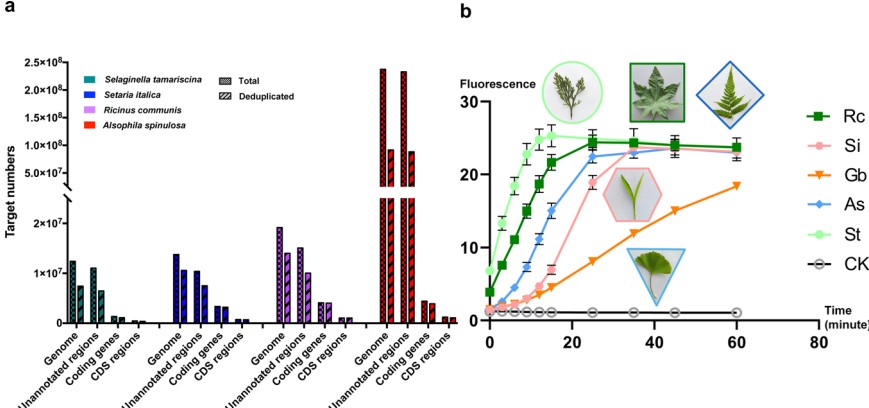

**Fig. 5 Application of GAGE in plants from different classes. a** Statistics of Targets in different classification derived from the four genomes. Genome: Targets located in the whole genome; Unannotated regions: Targets located in the unannotated regions of genome; Coding genes: Targets located in coding genes; CDS: Targets located in CDS. **b** Result of employing GAGE in plants from different classes. The DNA substrates and crRNA of every group were as follow: Rc: ITS2 fragments of *R. communis* + Rc_crRNA; Si: ITS2 fragments of *Set. italica* + Si_crRNA; Gb: ITS2 fragments of *G. biloba* + Gb_crRNA; As: ITS2 fragments of *A. spinulosa* + As_crRNA; St: ITS2 fragments of *Sel. tamariscina* + St_crRNA and CK (negative control): nuclease-free water + Cs_crRNA. All plots data represent means +/− standard deviation (SD) from three independent replicates.

performed using high-copy Rc_target3, Si_target3, As_target2, Gb_target1, and St_target1 as targets, and the matched crRNAs (Rc_crRNA, Si_crRNA, Gb_crRNA, As_crRNA, and St_crRNA) were synthesized based on them. The ITS2 fragments of the above plants were used as DNA substrates for their respective assays. The results (Fig. 5b) showed that each group had a significant fluorophore signal, which confirmed that GAGE has great universality in different plant classes, including angiosperms, gymnosperms, ferns, and lycophytes.

## Discussion
The number of plant species on earth remains highly controversial and speculative, whereas it is clear that the identification of the enormous number plants is a challenging and long-term task[23]. The whole genome can be exploited for plant species identification, and the technological advancements are making it possible. As the pioneer of the whole genome identification, GAGE takes advantage of both of genome analysis and genome editing, providing a new insight into plant species identification.

The major advantage of GAGE is embodied in the full utilization of the whole genome through bioinformatic analysis, chiefly appearing in the following areas: (i) GAGE searches and detects target sequences with PAM from the whole genome rather

than the specific regions. Although there are huge differences across the plants studied, there is a potential target sequence per 18–30 bp in the genome on average. Given the large size of the genomes, there are numerous target sequences, which offer enough information for plant species identification and enormous potential for GAGE. (ii) The selected target sequence in the subject plant is further compared to the whole genomes of its adulterants, which show the maintenance of the specificity of the target sequence and the accuracy of GAGE at the basal level. Meanwhile, the predicted off targets can also be analyzed to overcome the disadvantage of off target mutations in CRISPR/Cas12a system. (iii) The number of the published genomes, at the time of this experiment there was more than 700, is increasing rapidly each year[10]. Therefore, a huge number of published genomes provide a firm database and great potential for GAGE.

Another advantage of GAGE is the inclusion of a wide variety of outcomes from the previous studies. First, this study used the target sequences from the ITS2 region as objects to establish the method of GAGE. The ITS2 region of *Cr. sativus* had been demonstrated to have highly intraspecific conservatism and species-specificity. So this selected target sequence with PAM unique to the adulterants of *Cr. sativus* lets this ITS2 region become the ideal object in GAGE, and the experimental results

showed its high efficiency. Thereby, the ITS2 region of *Cr. sativus* to GAGE, as *Pisum sativum* (pea) to Mendelian, cannot be missed or replaced.

Second, other eligible regions also would be useful outside of the ITS2 region. For example, Kittisak Buddhachat et al. successfully used *trnL* region to identify *Phyllanthus* species[24], which provided support for the identification of closely related species. Moreover, the universal plant barcode regions such as *rbcL* + *matK*[6], the other genes containing high intraspecific conservatism and species-specificity like ycf1 and *leafy*[19,20] should be focused on during the selection of Targets.

Third, GAGE has great scalability, and the introduction of other detection systems would extend the application scope of GAGE. For example, both CRISPR/Cas13a and CRISPR/Cas12b systems have different PAMs compared to the CRISPR/Cas12a system[25,26], which would expand the Targets library and further develop the potential of the whole genome for plant species identification. Moreover, PAMs can be added to the target sequence by PCR, which would solve the restriction of PAMs and give the ability of detecting random sequences to GAGE. Previous studies on the regions used to identify plant species provided the theoretical supportive basis for the genome analysis (GA) and the novel detection methods shed new light on the advancements of genome editing (GE).

The superiority of GAGE is mainly centered on the genome analysis and genome editing. The bioinformatic analyses for identification, including the construction of Targets libraries, species-specificity analysis of targets, off target prediction, and designation of crRNA are possible to be integrated into a software with a graphical user interface (GUI), which may greatly simply the procedures and increase the analysis capability. The procedures of GAGE would be simplified by integrating the preparation of DNA substrate and detection, such as recombinase polymerase amplification (RPA)[27]. In addition, the application of genomic DNA without amplification and purification in GAGE was considered as a great advantage and is worth investigating. Notably, GAGE is not just restricted in a plant field, but it can also be applied to many other fields such as the species identification of animals and fungus and the detection of specific genes in molecular cloning experiment as well as transgenic identification.

As a pioneering identification method based on the whole genome, undoubtedly, GAGE has limitations in some aspects now. One limiting factor is the number of published genomes of medicinal plants remains small compared to the total amount of plant species. However, the cost of genome sequencing continues to fall, driven by ongoing innovation in sequencing technology, and the number of published genomes is increasing year by year. It is foreseeable that in the near future the number of published genomes will no longer be a limiting factor to GAGE. The other limitation is that the analysis of genomes requires simplification, which will be accomplished with the evolution of computer hardware and software.

The main use cases of GAGE consist of the following points. GAGE is effective when used with a boarder range of plant species, and we demonstrated that GAGE worked well with plant species from different classes including angiosperms, gymnosperms, ferns, and lycophytes. Based on these findings, we presume that GAGE is also suitable for the plant species from other classes. The second use is the identification of plant species with high economic and medicinal value. We performed GAGE with the identification of saffron and its adulterants, and the results showed that GAGE had great potential for the precise identification for other plant species with high economic and medicinal value. In addition, GAGE is useful for the identification of closely related plant species. Certainly, there are a lot of differences in genome sequences between different plant species and GAGE can

screen a unique target sequence based on these differences by aligning the genome sequences of different plant species. Thus, GAGE has potential applications in the identification of closely related species, which will attract great attention.

Carolus Linnaeus developed the binomial system 268 years ago and Charles Darwin published his famous theory of evolution 162 years ago. Both of them revolutionized the plant taxonomy[3]. Today, we present GAGE, a convenient, quick and highly sensitive method by which we are able to identify any plant species with published genome in principle, demystifying a tip of the iceberg for era of whole genome-based species identification.

## Methods

**Collection of plant source**. Leaves, stigmas or young buds were collected from plant samples and frozen in liquid nitrogen and stored at −80 °C. All of the fresh plants were identified by Professor Yulin Lin from the Institute of Medicinal Plant Development, Chinese Academy of Medical Sciences. The classes, families, origins and types are described in Table S4.

**Construction of Targets library**. The genomes of *Ricinus communis* (GCF_000151685.1), *Setaria italica* (GCF_000263155.2) and *Selaginella tamariscina* (GCA_003024785.1) were downloaded from NCBI database (https://www.ncbi.nlm.nih.gov) and the genomes of *Ginkgo biloba* (GWHBAVD00000000) was downloaded from NGDC database (https://ngdc.cncb.ac.cn/). The genome of *Alsophila spinulosa* was provided by Professor Quanzi Li from the Chinese Academy of Forestry and could be downloaded from Figshare (https://doi.org/10.6084/m9.figshare.19075346)[28]. As for *Crocus sativus*, the unpublished genome was obtained from our team. The genomes (L = genome length) of all six species were cut into 25 bp fragments by Jellyfish (v1.1.12) to generate (L-25 + 1) 25-mers with the copy number using the default parameters. The 25-mers with PAM (TTTV starting or VAAA ending) sequences were extracted and compared to their genomes, respectively, by Bowtie (v1.1.0) with the default parameters. The crRNAs were designed based on the selected Targets according to the manufacture' instructions[29,30].

**Species specificity analysis of Targets in *Crocus sativus***. The genomes of the three adulterants, *Ca. tinctorius* (GCA_001633085.1), *Nelumbo nucifera* (GCF_000365185.1), and *Zea mays* (GCF_902167145.1) were downloaded from the NCBI database. The Targets in ITS2 region were extracted and mapped to the genome of *Cr. sativus* and its three adulterants by Cas-OFFinder (v2.4).

**DNA extraction, amplification, and purification**. The total DNA extractions were performed utilizing the Plant Genomic DNA Kit (Tiangen Biotech (Beijing) Co. Ltd., China) according to the manufacturer's instructions. The DNA concentration and quality were assessed by Nanodrop 2000C spectrophotometry (Thermo Fisher Scientific Inc., China) and 0.8% (w/v) agarose gel electrophoresis at 120 V for 40 min (Bio Rad Laboratories Inc., USA), respectively. According to the procedure provided by Chen et al.[31], PCR amplification of ITS2 was performed in 50 μL reaction mixtures using 2 × Taq MasterMix (AidLab Biotechnologies Co. Ltd., China), with the annealing temperature increased to 58 °C for 40 cycles. The PCR products were purified by Universal DNA Purification Kit (Tiangen Biotech (Beijing) Co. Ltd., China) and were quantified and assessed by Nanodrop 2000C spectrophotometry and 2% (w/v) agarose gel electrophoresis at 120 V for 30 min. Resulting DNA products were used in the following assay.

**Cas12a endonuclease activity and collateral cleavage activity**. The endonuclease and collateral cleavage activity of LbaCas12a (New England Biolabs (Beijing) Ltd., China) were tested in 30 μL reaction mixtures according to the manufacturer's instructions. The reaction system was comprised of 3 μL 10× NEBuffer 2.1 (New England Biolabs (Beijing) Ltd., China), 20 μL nuclease-free water, 1 μL Cas12a (final concentration: 33 nM), 3 μL crRNA (final concentration: 300 nM), and 3 μL DNA substrate from the previous section (final concentration: 10 nM). The complex of Cas12a and crRNA were pre-incubated at room temperature for 10 min, then the DNA substrate was added, and the mixture was incubated at 37 °C for 3 h. Finally, after inactivating Cas12 at 65 °C for 10 min, the mixture was checked by 2% (w/v) agarose gel electrophoresis at 120 V for 30 min. The test of collateral cleavage activity was performed according to the procedures provided by Andrea Bonini et al.[16], in which a 50 bp single-stranded DNA was synthesized (Fig. S1, Genscript Co. Ltd., China). 3 μL of this 50 bp ssDNA (final concentration: 1 μM) was added to above reaction mixture together with the DNA substrate. The mixture was incubated at 37 °C for 1 h. The inactivating of Cas12a and agarose gel electrophoresis were the same as described above.

**Fluorophore quencher-labeled reporter detection**. The fluorophore quencher (FQ)-labeled reporter detection was performed in 100 μL reaction mixtures, using 10 μL 10× NEBuffer 2.1, 2 μL Cas12a (final concentration: 20 nM), 3 μL crRNA

(final concentration: 300 nM), 10 μL DNA substrate (final concentration: 1 ng/μL), 4 μL Poly_A_FQ (Genscript Co. Ltd., China, final concentration: 400 nM), and 71 μL nuclease-free water. The complex of Cas12a and crRNA was pre-incubated for 10 min at room temperature, then substrate DNA and Poly_A_FQ were added. Finally, the reaction mixture was incubated at 37 °C and detected at 0, 3, 6, 9, 12, 15, 25, 35, 45, and 60 min, respectively. The visual fluorescence signal was detected by BG-Vtrans520 (Baygene Biotech Co. Ltd.) The fluorescence intensity was read at $\lambda_{ex}$ 483 nm/$\lambda_{em}$ 535 nm by the fluorescence microplate analyzer (Thermo Scientific Inc., China).

**Statistics and reproducibility**. All statistical tests were performed using GraphPad Prism 9 software and each fluorophore quencher-labeled reporter detection was performed in triplicate as three technical replicates.

**Reporting summary**. Further information on research design is available in the Nature Research Reporting Summary linked to this article.

## Data availability
The source data for the graphs are shown in "Supplementary Data 1". The genome source data used in the genome analysis is described in the "Methods" section in detail. The raw data of genome sequencing of *Crocus sativus* have been deposited to the Genome Sequence Archive at the National Genomics Data Center under BioProject no. PRJCA010845.

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

## Acknowledgements
We are grateful to Q. Li for providing *Alsophila spinulosa* genome, C. Fan for providing *Carthamus tinctorius* and Y. Lin for identifying plants. This work was supported by National Nature Science Foundation of China (81874339) and Chinese Academy of Medical Sciences, Innovation Fund for Medical Sciences (CIFMS) (2021-I2M-1-022).

## Author contributions
J.S. conceived and designed the project. L.H., W.X., G.Q., and T.X. conducted experiments and analyzed results. W.X. and Z.X. performed bioinformatics analysis. L.H. and W.X. wrote this manuscript. J.S. and H.L. revised this manuscript.

## Competing interests
The authors disclose an International Patent Application (PCT/CN2021/138005) relating to the aspects of this work has been filed under Patent Cooperation Treaty.
