## [Peer Review File · Communications Biology]

Reviewers' comments:

Reviewer #1 (Remarks to the Author):

A new identification method for plant species basing on whole genome analysis and genome editing have been proposed in the manuscript, which is attractive and meaningful.

However, there are still some suggestions needed to be considered as below:

1.The describe of "similar to how Mendel chose pea" is not rigorous, the reason you choose saffron to establish GAGE could be described in detail.

2.The adulterants of *Carthamus tinctorius* are all from the other genus. In many cases, closely related species are often used as adulterants because of their similarity in appearance. Whether a good identification effect can still be obtained, when several species of the same genus are used as materials for adulteration identification.

3.In this paper, the same system of ITS2 fragment based on saffron and its adulterated products was used for identification using GAGE. However, if the saffron is randomly purchased from the market, can the GAGE method identify adulteration in it?

4.Using GAGE for species identification, whether it has the characteristics of easy operation and widely used like DNA barcoding technology?

5.At present, the whole genomes of no more than 150 medicinal plants have been published, and the whole genomes of many important medicinal plants have not been published. The identification of adulteration generally refers to the adulteration of medicinal plants or edible plants. Whether GAGE has limitations in the identification of adulteration.

Reviewer #2 (Remarks to the Author):

This manuscript presents a Crisper/Cas12a - based method for identifying plant species. My primary concern is that I think this system's limitations and use cases are not adequately discussed. Given the short DNA sequences used for identification, the proposed system will only work within small sets of species, similar to the ancient PCR amplicon-restriction enzyme-based approach. Also, I think the manuscript is unclear in many places, as detailed in the comments below.

Detailed comments:

Figure 1: Please spell out abbreviations in legend, such as FAM - BHQ, RuvC.

The lower right part of Figure 1: It's unclear how fluorescence detection works. From the figure, I see that RuvC cuts DNA - does RuvC also cut FAM - BHQ? Or how is the blue FAM transformed into yellow FAM?

p. 4, line 62: What's the definition of a target sequence?

p. 4, line 63: What's an adjacent protospacer motif? Maybe a few words in the Introduction on how Crispr/Cas12a works would be beneficial.

p. 6, lines 100 - 103. '... a 50 bp ssDNA was digested...', but Figure 3B lane B2 does show a ~50 bp band. Which would mean the 50 bp ssDNA wasn't digested?

p. 6, lines 103 - 105: I think a reference to a figure or image documenting the fluorescent signal is needed here.

p. 6, lines 106 - 107: What is 'two systems' referring to? The legend to Figure 3C doesn't explain either.

Figure 3C. What is represented by the circle?

p. 6, line 109: Instead of 'rose rapidly in a short time,' can you be more specific?

p. 6, line 110: '... the result provided a firm support for the specificity of GAGE.'. But that is using ITS2 only, not the entire genomic DNA? If yes, this statement should be restricted to the ITS2, and if this indeed signifies overall specificity across the entire genome, a reference should be provided.

p. 8, line 131: What's meant by 'adulterants'? After reading the sentence that follows (lines 131 - 134), I still don't understand.

p. 8, lines 134 - 135: I don't understand what this sentence means.

p. 8, line 143. OK, so only ITS2 is used for identification. How did you synthesize this sequence, by PCR?

p. 8. line 145: Instead of 'enormous', maybe detectible?

Figure 4B: As mentioned earlier, please explain the circle, square, and triangles - do they contain stamens? Why are they included?

pages 11 - 13, Discussion:

The authors didn't convince me that this new method for plant identification is practical or needed at this point. The short fragments don't guarantee species specificity and only work in a setting that involves a small number of species, as a PCR amplicon-restriction enzyme type of identification system. I think a more nuanced discussion of the advantages and disadvantages of GAGE, and the specific situations where it may be most helpful, would be beneficial.

Rebuttal Letter

Reviewers' comments:

Reviewer #1 (Remarks to the Author):

A new identification method for plant species basing on whole genome analysis and genome editing have been proposed in the manuscript, which is attractive and meaningful.

Response: Thank you very much for your positive comments!

However, there are still some suggestions needed to be considered as below:

1.The describe of “similar to how Mendel chose pea” is not rigorous, the reason you choose saffron to establish GAGE could be described in detail.

Response: We are grateful to you for your encouraging comments! We have added the detailed reason why we chose saffron in line 74 of page 5. The obvious variation of character in pea is essential for the study on the genetic law and whole genome sequences are necessary for our GAGE. We sequenced the whole genome of saffron and obtained a reference genome, which provided a firm foundation for genome analysis. Moreover, saffron is a well-known and medicinal material, the identification of it from adulterants has received great attention and has implications for clinical applications. Selection of the first subject is essential for the establishment of a new approach. We selected saffron and successfully established GAGE with it, just like Mendel chose pea to investigate the genetic law.

2.The adulterants of *Carthamus tinctorius* are all from the other genus. In many cases, closely related species are often used as adulterants because of their similarity in appearance. Whether a good identification effect can still be obtained, when several species of the same genus are used as materials for adulteration identification.

Response: You are right. Closely related species are often used as adulterants. However, saffron doesn't have adulterants from the same genus according to reference^[1], so we chose the common adulterants from other genera.

For closely related species, certainly, there are a lot of differences across their genomes. Through bioinformatic analysis (Genome Analysis), we can search the target sequences that reflected these differences and further verify them by experimental evidence (Genome Editing).

As described, the candidate target sequences were ubiquitously distributed in the whole genome and the target sequences were selected from at least 10 million candidate target sequences, the specificity of which was further verified by mapping them to the whole genomes of closely related species. Only the target sequences present in the genome of plant species and not in the genomes of closely related species were selected and considered for use. Thus, the specificity of short DNA sequences was fully guaranteed by bioinformatic analysis, and it was further demonstrated in the following experimental evidence.

In conclusion, it is available to search an appropriate target sequence for each plant species and we believe that GAGE applies to the identification of closely related species.

[1] Shi, R. et al. Hierarchical nanostructuring array enhances mid-hybridization for accurate herbal identification via ITS2 DNA barcode. *Anal. Chem.* 92, 2136-2144 (2020).

3.In this paper, the same system of ITS2 fragment based on saffron and its adulterated products was used for identification using GAGE. However, if the saffron is randomly purchased from the market, can the GAGE method identify adulteration in it?

Response: In our study, we directly used fresh saffron to establish GAGE.

Actually, we also performed GAGE with saffron from the market; ten samples were randomly selected and identified by GAGE at different times, and the results showed that all samples were saffron and no adulteration existed (Fig. 1). Meanwhile, all ten samples were identified based on the morphological characteristics and three of them were identified based on DNA barcoding, and the results showed that the ten samples were saffron (Fig. 2). Considering that our primary purpose is to establish a new plant species identification method, we didn't test more samples from the market and didn't display this work in our manuscript.

Fig. 1 Identification of ten randomly selected samples of saffron from market with GAGE. Using Cs_crRNA as crRNA to identify saffron, every group had the same composition except the DNA substrate, including Cas12a, Cs_crRNA, NEBuffer 2.1, ssDNA reporter and H₂O. The DNA substrates of Cs1 to Cs10 were the ITS2 fragments of *Cr. Sativus*, which were from ten randomly selected saffrons. In CK (negative control), we used H₂O to replace the DNA substrate.

```

Cs   CGCCTCCCGTCGCTCCCCACAGCCGTGCGGATGCGGAGATTGGCCCCCGCTGCTCCGTGCGCGGGGGTC
Cs1  .....
Cs3  .....
Cs10 .....

Cs   GAAGTGCCGGCCGTCGTCGGGCCTGGCGCGGCGAATGGTGGACGAATACATCGTTGTTTGTGCCTACTCT
Cs1  .....
Cs3  .....
Cs10 .....

Cs   CCGTGTCCCTGCCCCTCAACAATGCGACATGTCGTCGTCGGACCCCTCACCATGGACCCTTCCGGCTCCG
Cs1  .....
Cs3  .....
Cs10 .....

Cs   AATAAGAAGAAGGAACCGTCCTCGGAACG
Cs1  .....
Cs3  .....
Cs10 .....

```

Fig. 2 Alignment of ITS2 fragments from the three selected saffrons.

Cs is the standard sequence of ITS2 of *Cr. Sativus*^[2], Cs1, Cs3 and Cs10 are the sequences of DNA substrates of Cs1, Cs3 and Cs10 in Fig.1, respectively. The dot means the base is identical to the one in Cs.

If there are some adulterants in the saffron samples from the market, we believe that GAGE can identify them. As noted, GAGE can identify any plant species with a designed crRNA. We proved that GAGE can identify saffron with high sensitivity and applied GAGE to five plant species from different classes with target sequences. Similarly, a species-specific crRNA will be designed and used to precisely identify the adulterants of saffron.

[2] Chen S. et al. (2015). *Standard DNA barcodes of Chinese materia medica in Chinese pharmacopoeia*. Beijing: Science Press.

4.Using GAGE for species identification, whether it has the characteristics of easy operation and widely used like DNA barcoding technology?

Response: Yes, GAGE has several convenient characteristics, such as simple operation, high efficiency and a wide range of application. Similar to DNA barcoding, GAGE has few ingredients and needs only incubation and detection. GAGE differs from DNA barcoding in the identification – GAGE depends on

visual results of fluorescence instead of sequencing analysis, and this means GAGE does not require professional operators and instruments, which greatly expands its application scope.

5. At present, the whole genomes of no more than 150 medicinal plants have been published, and the whole genomes of many important medicinal plants have not been published. The identification of adulteration generally refers to the adulteration of medicinal plants or edible plants. Whether GAGE has limitations in the identification of adulteration.

Response: This is a great point! We added a discussion of the limitations in the "Discussion" section. As a pioneering identification method based on the whole genome, undoubtedly, GAGE has limitations in some aspects. The number of published genomes of medicinal plants remains small. However, the cost of genome sequencing continues to fall, driven by ongoing innovation in sequencing technology. According to statistics, there are 72 published medicinal plant genomes in 2020, which is over three times more than that in 2019^[2]. It is foreseeable that the number of published genomes will no longer be a limiting factor to GAGE in the near future. In addition to the small number of published genomes, the strategies of genome analysis can be further optimized to select candidate target sequences without the restriction of PAM, which will extend the screening scope of target sequences for the identification of closely related adulterants.

[3] He, S. et al. MPOD: Applications of integrated multi-omics database for medicinal plants. Plant Biotechnol. J. (2021).

Reviewer #2 (Remarks to the Author):

This manuscript presents a Crisper/Cas12a - based method for identifying plant species. My primary concern is that I think this system's limitations and use cases are not adequately discussed. Given the short DNA sequences used for identification, the proposed system will only work within small sets of species, similar to the ancient PCR amplicon-restriction enzyme-based approach. Also, I think the manuscript is unclear in many places, as detailed in the comments below.

Response: We are sorry that we didn't describe GAGE clearly. Actually, our GAGE includes two key steps: bioinformatic analysis and experimental evidence, i.e., genome analysis (GA) and genome editing (GE). Through bioinformatic analysis, we screened candidate target sequences with a protospacer adjacent motif (PAM) from the whole genome of plant species, and further selected target sequences which do not exist in the related species by mapping it to the genomes of the related species. According to target sequence, a crRNA was designed and used to generate the experiment evidence. The crRNA specifically recognized and bound to the target sequence in the DNA substrate from plant species, which activated the collateral cleavage activity of Cas12 and drove the generation of fluorescence. On the contrary, there was no fluorescence signal with the DNA substrate from the related species due to the lack of target sequence. Therefore, we added three red lines to mark and distinguish Genome Analysis and Genome Editing (Fig.1).

According to your comments, we discussed the limitations and use cases in line 286 of page 15. The main limitations of GAGE consist of the following aspects: One is that the small number of published genomes restricts the current application of GAGE, but it is foreseeable that the number of published genomes will no longer be a limiting factor to GAGE in the near future because of the increasing amount of published genomes^[1, 2]. The other is that the analysis of genomes remains to be simplified and integrated, which will be accomplished with the continuous evolution of computer hardware and software.

Considering use cases, we added the following three points: The first one is the application in a boarder range of plant species. We proved that GAGE applied to the plant species from different classes including angiosperms, gymnosperms, ferns, and lycophytes. With this, we presume that GAGE is also suitable for the plant species from other classes. The second use case is the identification of plant species with high economic and medicinal value. We performed GAGE with the identification of saffron and its adulterants, and the results proved that GAGE has great potential for the precise identification for other plant species with high economic and medicinal value. The final consideration is the identification of closely related plant species. Certainly, there are a lot of differences in genome sequences between different plant species and GAGE can screen a unique target sequence based on these differences by aligning the genome sequences of different plant species. Thus, GAGE has potential applications in the identification of closely related species, which will attract great attention.

In general, it is believed that short DNA sequence has limited species specificity. However, in our study, benefiting from the genome analysis, the target sequence in GAGE has extremely high species specificity. As described, the target sequence was selected from over 59 million candidate target sequences, the specificity of which was further verified by mapping it to the whole genomes of related species. The target sequences present only in the genome of saffron and not in the genomes of its adulterants were selected and considered for use. Thus, the specificity of short DNA sequences was fully guaranteed by the bioinformatic analysis. The specificity was further validated in the following experimental evidence. The target sequence in the ITS2 region was used as a case to establish GAGE, meanwhile, other appropriate target sequences can also be used for the identification of saffron. Most importantly, the strategy of GAGE is also applicable to the identification of other plant species. Consequently, as the first plant species identification method based on the whole genome analysis, GAGE is completely different from the PCR amplicon-restriction enzyme-based

approach.

Following your advice, we had revised our manuscript and provided more information.

[1] Human genome at ten: The sequence explosion. Nature. 464, 670-1 (2010).

[2] Kersey PJ. Plant genome sequences: past, present, future. Curr. Opin. Plant Biol. 48,1-8 (2019).

Fig.1 The strategy of the whole Genome Analysis and Genome Editing (GAGE)

Detailed comments:

Figure 1: Please spell out abbreviations in legend, such as FAM - BHQ, RuvC.

The lower right part of Figure 1: It's unclear how fluorescence detection works. From the Figure, I see that RuvC cuts DNA - does RuvC also cut FAM - BHQ? Or how is the blue FAM transformed into yellow FAM?

Response: Thanks for your advice! We have added all full names in the legend of Figure 1. Noted, RuvC means a RuvC-like nuclease domain in Cas12a, which is named because of the similar structure to the RuvC enzyme in *Escherichia coli*. The Ruv enzymes received their names due to their association with the resistance to ultraviolet light (Ruv) and don't have a full name^[3]. In addition, a detailed description of Figure 1 and a description of how GAGE works was added in line 53 of page 3.

In the fluorescence quenching experiment, the RuvC domain of Cas12 indeed cuts FAM - BHQ (FAM - AAAAAAAAAA - BHQ). The fluorophore (FAM) is linked to quencher (BHQ) through ten dAs and is initially quenched, so it has dark (blue) color. When the collateral cleavage activity of Cas12 is activated, the single sequence DNA (ten dAs) between FAM and BHQ is cut by the RuvC domain and FAM breaks away from BHQ, which absorbs excitation light and fluoresces. Thus, the color of FAM transforms from blue to yellow.

[3] Otsuji N. et al. Isolation and characterization of an *Escherichia coli* ruv mutant which forms nonseptate filaments after low doses of ultraviolet light irradiation. *J. Bacteriol.* 117, 337-44 (1974).

p. 4, line 62: What's the definition of a target sequence?

Response: Thank you for prompting this clarification. We added a description of target sequence in line 55 of page 3. The target sequence is a genome-derived, selected, species-specific, 25 bp sequence with a protospacer adjacent motif (TTTV) used for identification. As mentioned, we first screened the sequences with PAM, which were named "candidate target sequences". Candidate target sequences were further selected by mapping them to the genomes of the related species. The final selected target sequence don't exist in the related species.

In GAGE, crRNAs were designed based on target sequences. Moreover, target sequences were recognized and bound by the matched crRNAs, which drove the identification of plant species.

In line 80 of page 5, the correct expression should be "candidate target sequences", which would be further screened by mapping to the genomes of adulterants. After that, the species-specific sequences used for identification were named "target sequences".

p. 4, line 63: What's an adjacent protospacer motif? Maybe a few words in the Introduction on how Crispr/Cas12a works would be beneficial.

Response: Thank you for this suggestion and it is a good advice to introduce how CRISPR/Cas12a works in detail. We added an introduction of the CRISPR/Cas12a system in line 53 of page 3. The protospacer adjacent motif (PAM) is a short DNA sequence (usually 2-6 base pairs in length) immediately following the DNA sequence targeted by the Cas nuclease in the CRISPR system, which is necessary for a Cas nuclease to bind and cleave the target DNA sequence^[4].

[4] Takashi, Y. et al. Structural Basis for the Canonical and Non-canonical PAM Recognition by CRISPR-Cpf1. *Mol. Cell.* 64, 633-645 (2017).

p. 6, lines 100 - 103. '... a 50 bp ssDNA was digested...', but Figure 3B lane B2 does show a ~50 bp band. Which would mean the 50 bp ssDNA wasn't digested?

Response: Thank you for your careful review! We revised the description of Figure 3 in line 118 of page 7 and added a marker for 50 bp in Figure 3B.

In Figure 3B, we used a 50 bp ssDNA to verify the collateral cleavage activity of Cas12a. The lowest three bands in lane B1, B2, and B3 were the 50 bp ssDNA, the sequences of which are shown in Figure S1. The lane B1, B2 and B3 had the same 50 bp ssDNA, but only the ssDNA in lane B1 was digested by Cas12 coupled with substrate DNA and crRNA. As shown in Figure 3B, the band of lane B1 was more dim than that of lanes B2 and B3.

p. 6, lines 103 - 105: I think a reference to a Figure or image documenting the fluorescent signal is needed here.

Response: Thank you for your advice! An image of visual fluorescent signal was added as Figure 3C.

Fig. 3 Feasibility, specificity and sensitivity of GAGE

p. 6, lines 106 - 107: What is 'two systems' referring to? The legend to Figure 3C

doesn't explain either.

Response: Thanks for your comment! Two systems referred to the two sets of experiments that we performed to verify the specificity of GAGE. For clarity, we replaced “system” with “group”. GAGE has six ingredients including buffer, Cas12a, crRNA, H₂O, DNA substrate, and ssDNA reporter. In Figure 3D, the two groups had the same components except DNA substrate. Cs used the ITS2 fragments of *Cr. Sativus* as DNA substrate and CK (negative control) used H₂O instead of DNA substrate.

Figure 3C. What is represented by the circle?

Response: The plant materials in the circle are the stigma of saffron. To directly show the original plant species of DNA substrate in each experiment, we added plant photographs in the result of fluorescence, the shapes and colors of which were matched to the right legend. An introduction of plant photographs has been added into the legend in Figure 3D.

p. 6, line 110: '... the result provided a firm support for the specificity of GAGE.'. But that is using ITS2 only, not the entire genomic DNA? If yes, this statement should be restricted to the ITS2, and if this indeed signifies overall specificity across the entire genome, a reference should be provided.

Response: As you said, we amplified the regions containing target sequence by PCR, which were used as DNA substrate. But we analyzed the specificity of target sequence by mapping it to the genomes of adulterants and chose the species-specific target sequence for identification. In other words, the specificity of target sequence was guaranteed by bioinformatic analysis (GA), and it was further verified in the following experimental evidence (GE). The target sequence in ITS2 region was used as a case to establish GAGE, meanwhile, other appropriate target sequences can also be used for the identification of saffron. If the target sequence locates at other regions of genomic DNA, these regions will

be amplified and used as DNA substrate.

p. 8, line 131: What's meant by 'adulterants'? After reading the sentence that follows (lines 131 - 134), I still don't understand.

Response: Thank you for your prompt for clarification. We added a detailed introduction of adulterants in line 155 of page 9. The stigma of the saffron is used as colorants, medicines, and fragrances. Because of its high value and low yield, many plant materials of other plant species with similar characteristics, including the stigma of saffron, the flower of safflower, the stamen of lotus, and the style and stigma of corn, are dyed red and used to impersonate the stigma of the saffron.

p. 8, lines 134 - 135: I don't understand what this sentence means.

Response: We added a more detailed description of how we chose the target sequences in line 160 of page 9. Only target sequences present in the genome of saffron and not in the genomes of its adulterants were considered for use. The crRNA was designed according to the target sequence, and would recognize and bind to the target sequence in the DNA substrate from saffron, which further drove the generation of fluorescence. Because there was no sequence in the genomes of adulterants that matched the designed crRNA, no fluorescence would be generated when the DNA substrates from adulterants were used.

p. 8, line 143. OK, so only ITS2 is used for identification. How did you synthesize this sequence, by PCR?

Response: Yes, the ITS2 sequence was obtained by PCR.

p. 8. line 145: Instead of 'enormous', maybe detectible?

Response: Thank you for your suggestion. We replaced “enormous” with “detectable” in line 176 of page 10.

Figure 4B: As mentioned earlier, please explain the circle, square, and triangles - do they contain stamens? Why are they included?

Response: We added an introduction of plant photographs in the legend of Figure 4B. The different shapes and colors only represent different kinds of plant species without any other implication. The plant materials in the circle, square, triangle and inverse triangle are the stigma of saffron, the flower of safflower, the stamen of lotus, and the style and stigma of corn, respectively. Not all of them contain stamens. As described above, the adulterants of saffron refer to the dyed plant materials. We added these plant photographs to directly show the original plant species of DNA substrate in each experiment.

pages 11 - 13, Discussion:

The authors didn't convince me that this new method for plant identification is practical or needed at this point. The short fragments don't guarantee species specificity and only work in a setting that involves a small number of species, as a PCR amplicon-restriction enzyme type of identification system. I think a more nuanced discussion of the advantages and disadvantages of GAGE, and the specific situations where it may be most helpful, would be beneficial.

Response: We apologize that we did not clearly describe our GAGE. GAGE is the first identification method based on the whole genome analysis and has the potential to identify any plant species with the whole genome.

The unusual short fragments (target sequences) in GAGE have extremely high species specificity. As noted, the target sequence was selected from over 59 million candidate target sequences, the specificity of which was verified by mapping it to the whole genomes of related species. Only target sequences present in the genome of saffron and not in the genomes of its adulterants were selected and considered for use. Thus, the specificity of short fragments was fully guaranteed. After that, it was further validated by experimental evidence. Accordingly, we believe that GAGE applies to the identification of a wide range of plant species.

We discussed the advantages, disadvantages, and the use cases of GAGE in line 286 of page 15. The advantages of GAGE are mainly concentrated in the utilization of the whole genome and flexible detection method. First, GAGE can identify any plant species at a species level based on the abundant information of the whole genome. Second, GAGE makes full use of previous studies to provide a reference for the genome analysis. Third, the simple workflow and visual result make GAGE broadly applicable. Moreover, GAEG has great scalability and other detection methods are also available to extend the application scope. For example, CRISPR/Cas13a and CRISPR/Cas12b systems have different PAMs compared to the CRISPR/Cas12a system, which can expand the screening scope of target sequences and further develop the potential of the whole genome for plant species identification.

The main disadvantages of GAGE includes the following two aspects. One is the limitation of small number of published genomes, but we expect this to no longer be a problem in the near future. The other is that the analysis of genomes remains to be simplified and integrated with advances in hardware and software.

The use cases of GAGE mainly include the following three points: The first one is the application in a boarder range of plant species; the second use case is the identification of plant species with high economic and medicinal value; the last one is the identification of closely related plant species. Consequently, as the first plant species identification method based on the whole genome analysis, GAGE is completely different from the PCR amplicon-restriction enzyme-based approach.

GAGE is the first plant species identification method based on the whole genome analysis, and it will play an important role in multiple identification areas and provide new insights into the application of the whole genome in species identification.

REVIEWERS' COMMENTS:

Reviewer #1 (Remarks to the Author):

The author has made a good answer and modifications to the previous questions. The applicability and limitations of GAGE are explained and supplemented. But here are some details that the author needs to check carefully.

For example, 1) P. 10, Fig 4, the circle of *C. sativus* is a little bit smaller and the red line is not very clear and obvious. 2) P. 10, line 186, "were as follow" should be revised as "were as follows". 3) Some words should not be italicized. For example, in lines 171 to 173; line 204: Table S3. 4) P 12, Fig. 5, the font for the legend in the figure could be larger.

Rebuttal Letter

REVIEWERS' COMMENTS:

Reviewer #1 (Remarks to the Author):

The author has made a good answer and modifications to the previous questions. The applicability and limitations of GAGE are explained and supplemented. But here are some details that the author needs to check carefully.

For example, 1) P. 10, Fig 4, the circle of *C. sativus* is a little bit smaller and the red line is not very clear and obvious. 2) P. 10, line 186, “were as follow” should be revised as “were as follows” . 3) Some words should not be italicized. For example, in lines 171 to 173; line 204: Table S3. 4) P 12, Fig. 5, the font for the legend in the figure could be larger.

Response: We are sincerely grateful to you for your constructive and positive comments. According to the raised points, we have carefully revised our manuscript.

For example, 1) P. 10, Fig 4, the circle of *C. sativus* is a little bit smaller and the red line is not very clear and obvious.

Response: We have enlarged the circle of *C. sativus* to the appropriate size and made the red line bold in Fig. 4.

Fig. 4 Feasibility, specificity and sensitivity of GAGE

2)P. 10, line 186, “were as follow” should be revised as “were as follows” .

Response: We have replaced "were as follow" with "were as follows" in line 454 of page 19.

3)Some words should not be italicized. For example, in lines 171 to 173; line 204: Table S3.

Response: Thanks for pointing it out! We conscientiously study the usage of italics. In our manuscript, all plants were represented with the names of biological species. The words mentioned above also belonged to the names of biological species. According to the format guidelines of this journal, the names of biological species should be italicized, which was further confirmed by consulting a recent article (DOI: <https://doi.org/10.1038/s42003-022-03523-5>). Taking into account the above situation, we didn't change it.

4) P 12, Fig. 5, the font for the legend in the figure could be larger.

Response: We have enlarged the font size to make the legend clearer.

A**B**
Fig. 5 Application of GAGE in plants from different classes